# Intelligent IoT Platform for Multiple PV Plant Monitoring

**DOI:** 10.3390/s23156674

**Published:** 2023-07-25

**Authors:** Ida Bagus Krishna Yoga Utama, Radityo Fajar Pamungkas, Muhammad Miftah Faridh, Yeong Min Jang

**Affiliations:** Department of Electronics Engineering, Kookmin University, Seoul 02707, Republic of Korea; idabaguskrishnayogautama@gmail.com (I.B.K.Y.U.); radityofajar@gmail.com (R.F.P.); muhammadmiftahfaridh@gmail.com (M.M.F.)

**Keywords:** photovoltaics, monitoring, IoT platform, prediction, anomaly detection

## Abstract

Due to the accelerated growth of the PV plant industry, multiple PV plants are being constructed in various locations. It is difficult to operate and maintain multiple PV plants in diverse locations. Consequently, a method for monitoring multiple PV plants on a single platform is required to satisfy the current industrial demand for monitoring multiple PV plants on a single platform. This work proposes a method to perform multiple PV plant monitoring using an IoT platform. Next-day power generation prediction and real-time anomaly detection are also proposed to enhance the developed IoT platform. From the results, an IoT platform is realized to monitor multiple PV plants, where the next day’s power generation prediction is made using five types of AI models, and an adaptive threshold isolation forest is utilized to perform sensor anomaly detection in each PV plant. Among five developed AI models for power generation prediction, BiLSTM became the best model with the best MSE, MAPE, MAE, and $R^{2}$ values of 0.0072, 0.1982, 0.0542, and 0.9664, respectively. Meanwhile, the proposed adaptive threshold isolation forest achieves the best performance when detecting anomalies in the sensor of the PV plant, with the highest precision of 0.9517.

## 1. Introduction

Rapid industrial and urban expansion necessitates more electricity to fuel the expansion. The majority of power plants use nonrenewable fuels such as coal, gas, and oil, making electricity generation the primary cause of air pollution. By producing energy from these resources, it severely damages the environment by polluting the air, water, and land [1]. To mitigate these issues and establish environmental sustainability, renewable energy sources are regarded as the most effective means of electricity generation. Numerous renewable energy sources can be used to generate electricity. Solar, wind, hydro, and geothermal energy are the most prevalent forms of renewable energy. Solar energy is the most promising of these options because sunlight is abundant, and solar radiation of approximately $1.73\times 10^{5}$ terawatts (TW) strikes the earth’s surface continuously [2].

In 2022, an additional 240 GW of PV plants were installed, and the total installed PV plant capacity in the world now reaches 1.2 TW [3]. It shows a huge demand for PV plants, and it is expected that PV plants installed capacity could reach up to 13.8 TW by 2040 [4]. The adoption of PV plants is also expected to rise as the U.S. Department of Energy issues the SunShot Initiative, which aims to half the price of solar energy by 2030 [5]. Due to the huge demand for PV, the development of PV technologies is also increasing rapidly. Numerous studies about PV have been conducted by many researchers around the world. Improving the efficiency of a PV panel has been conducted, and some researchers were able to produce a PV panel with an efficiency of up to 47%, a significant increase from the common PV panel which only has 15–20% efficiency [6].

PV plants are susceptible to malfunctions due to external and internal factors such as overcurrent, UV radiation, and human error. Numerous researchers have already explored the development of a fault-detection system for PV systems. Several studies concentrated on identifying PV panel faults by measuring the I–V current [7]. As described in [8,9], thermal photography is also used to detect defects in PV panels. Many studies about fault diagnosis in a PV plant focused on detecting faults and anomalies that occur in the PV plant equipment [7,8,9,10,11].

The PV plant is typically connected to the grid, where it can choose whether to transfer the energy to the grid or store it in an energy storage system (ESS). Choosing whether or not to trade energy involves a great deal of uncertainty. Predicting the next day’s PV energy production is crucial for decision making. By predicting the amount of energy that will be generated the following day, energy trading decisions will be simplified, whether the energy is kept or traded. When implementing next-day generated power prediction, it is possible to generate income projections because the quantity of energy to be traded is known in advance.

Numerous devices are installed in a PV plant, and each device must be consistently monitored to ensure plant operations. Monitoring has benefits in that the collected data are useful for PV fault detection and power generation predictions for the next day. Consequently, the monitoring system, defect detection, and prediction of the next day’s power generation should be performed simultaneously. Due to the enormous amount of equipment and data produced by a PV facility, it is impossible to perform this task manually. The IoT platform is an emerging technology that can capture data from a vast array of sensors and transform it into more valuable information. As a result, an IoT platform can be employed for PV plant real-time monitoring, including fault detection and prediction of the next day’s power generation. Multiple researchers propose an IoT platform for PV plant monitoring [12,13,14,15,16,17,18]. However, they all consider monitoring a single PV plant with a single IoT platform.

Given the rapid development of PV plants, particularly in Korea, numerous businesses are interested in investing in the construction of PV plants. A company may own multiple PV plants in different regions, necessitating the development of a new IoT platform capable of monitoring multiple PV plants simultaneously. This work proposes a method for establishing an intelligent IoT platform that includes methods for predicting the next day’s power generation and detecting anomalies in real time. In conclusion, the following points are the contributions of this work:Intelligent IoT platform architecture to monitor multiple PV plants.Method to perform AI-based next-day power generation prediction in multiple PV plants.Adaptive threshold Isolation Forest for detecting sensor malfunctions in multiple PV plants.

The rest of this paper is structured as follows: Section 2 presents a literature review of related topics in this work. Section 3 explains the methodology of the techniques presented in this work. Section 4 provides the detailed experiment results and discussion. Section 5 concludes this work.

## 2. Literature Review

This section discusses recent research in PV monitoring technologies, PV power generation prediction, and communication system anomaly detection. Numerous researchers around the globe have conducted extensive research on PV monitoring, but all of their findings are centered around monitoring a single PV plant. The Intelligent Monitoring System (IMS) is a low-cost, uncomplicated PV plant monitoring system developed by Masoud et al. Reference [12] used an IoT platform, a cloud service, and online monitoring software. An ensemble LSTM neural network predicts daily PV system output power in the IMS. In PV plant fault detection, ensembles of KNN, NB, and SVM are utilized to discover and identify problems. As described in [13], a Raspberry Pi can be used to monitor the operation of PV in a smart microgrid. The Raspberry Pi is connected to an Arduino in order to capture temperature data, with the data then being sent to Grafana for visualization.

Reference [14] proposed a low-cost supervisory control and data acquisition (SCADA) system for PV plant monitoring and local data recording. A Raspberry Pi, an Arduino, sensors, and serial communication cables are required to construct the low-cost SCADA system. For system and data flow monitoring, the open-source web-view platform Emoncms is employed. As proposed in [15], narrowband IoT (NB-IoT) communication technologies can be used to establish a PV monitoring system. They plan to acquire, store, and show PV monitoring data using the IoT platform Pycom and NB-IoT as the data transmission technologies.

Gonzalez et al. proposed an open-source IoT platform for PV plant I-V curve tracking. The proposed IoT platform uses open-source hardware and software; the Raspberry Pi is used for hardware. The platform then utilizes open-source software such as MariaDB, Python, and Grafana. Based on the results of the experiment, the I-V curve was effectively traced under real operating conditions [16].

To support the operation of the power system and ensure the safety and stability of the PV system, it is advantageous to have a reliable PV power generation prediction. Yao et al. propose a graph spatial-temporal attention neural network (GSTANN) for predicting the very short-term power generation [17]. The GSTANN model is trained using input from local meteorological data and satellite cloud images. The model is used for very short-term forecasting, with a range of 45 to 60 min. In comparison to other methods, such as ARIMA, SVR, and FC-LSTM, the proposed GSTANN model is able to produce the optimal model based on the results of the experiments.

Using historical weather data as input, a dual-stream CNN-LSTM (DSCLANet) model is employed to derive solar power predictions [18]. Comparing the DSCLANet to other models, such as LSTM-CNN, DenseNet, ELM, WPD-LSTM, and RCC-LSTM, the authors find that the DSCLANet obtains the best performance. Due to its dual architecture, the model’s complexity is high, despite its promising performance; it is intended to develop a single-architecture model.

Detecting anomalies assists in identifying sensor faults and reducing downtime in a PV plant. The utilization of Siamese-twin neural networks was proposed by Sajun et al. [19] for an anomaly detection system on edge devices within a PV plant. The proposed anomaly detection method attained an F1-score of 0.88. Its performance was assessed in multi-threading scenarios on edge devices, including the Raspberry Pi, Nvidia Jetson Nano, and Google Coral. Analysis of a single-edge device has the potential to accommodate 512 solar panels within a minute. Moreover, Ibrahim et al. [20] conducted a study to evaluate the efficacy of three established anomaly detection algorithms, namely Autoencoder LSTM (AE-LSTM), Facebook-Prophet, and isolation forest, in detecting anomalies on PV components. The assessment attained a maximum accuracy of 0.896.

## 3. Methodology

### 3.1. Intelligent IoT Platform Architecture

IoT technology is a network of connected devices where each device is connected over a communication network and exchanges data with each other. Generally, IoT devices are installed in a framework to enable easier deployment, control, and management. The framework is called the IoT Platform, which includes several tools and services to support IoT device operations. Typically, an IoT platform is composed of three layers: the device layer, the network layer, and the application layer. The device layer is the bottom layer, which contains all of the sensors or actuators required on the platform. The network layer is the middle layer, which holds the communication technology to enable data transport from the device layer to the upper layer. Many communication technologies and protocols are applied in the network layer, such as WiFi, cellular networks, or LoRa for communication technologies, and MQTT, HTTP, or WebSocket for communication protocols. Next, the application layer is the top layer, which collects, stores, and processes all transmitted data from the device layer. The application layer is composed of a cloud server that centrally manages the whole IoT platform.

A standard IoT platform cannot be used to monitor multiple PV facilities. Every minute, thousands of records are created in a single PV facility that consists of hundreds of PV panels and thousands of battery cells to store the energy produced. These thousands of data points are transmitted to the cloud server, and when the IoT platform manages multiple PV plants, the received data can reach millions of data points per minute. Handling a large quantity of data may place a strain on the server, resulting in decreased energy efficiency, increased latency, and reliability problems. In this paper, we introduce a new layer referred to as the “edge layer”, which utilizes an edge server as the link between the network layer and the application layer. The purpose of the edge server is to distribute the cloud server’s workload and outsource tasks. By distributing the processing burden to the edge server, it will reduce the cloud server’s load and the likelihood of a single point of failure, thereby increasing the platform’s reliability.

In this work, each monitored PV plant is equipped with an edge server. Each edge server only receives data from the particular PV plant it is located at, and the edge server collects and processes all data generated by the PV plant. By locating the edge server closer to the PV plant, latency is reduced and system response time is increased. This method enables distributed PV plant management and control. The data are stored in a data storage system on the edge server. The edge server then transmits the information to the cloud server. On the cloud server, all PV plant data are collected, stored, and used to train the intelligent system and provide the user with more accurate information.

Figure 1 depicts the IoT platform architecture proposed for monitoring multiple PV facilities. To accomplish its goals, the architecture can be broken down into several major components. The following is an explanation for each component:

#### 3.1.1. Message Queuing

The IoT platform includes a message queuing system. Message queuing is required to ensure that the transmitted data from the preceding layer is effectively retrieved at the destination, as the platform receives thousands of messages per minute. The message queue for receiving data from each PV plant is located on the cloud server. Message queuing telecommunication technology (MQTT) is used as the message queuing system in this work. MQTT uses the publish-subscribe mechanism to communicate data using a *topic* that hosts data exchanges between the publisher and the subscriber. Using MQTT, each edge server transmits data to a *topic* that is distinct based on data type and PV plant site identifier.

#### 3.1.2. Data Storage

All data collected from the PV plant is stored in a database system. The database is installed on edge and cloud servers. The database on the edge server only stores data from a PV plant. Meanwhile, the database on the cloud server accumulates the data from all PV plants. MariaDB is employed as the database in this work, where it utilizes a relational database management system (RDBMS) to improve efficiency when managing the stored data.

#### 3.1.3. Intelligent Services

The application of intelligent services facilitates the administration and maintenance of the platform. Two services are available on the platform. First, the AI-based power generation prediction, which uses weather data to predict the next day’s generated power. Predictions of electricity generation are essential for decision making in power trading. The prediction information facilitates the decision of whether to transfer the energy storage system’s (ESS) power to the grid or to retain the power in the ESS. The power generation prediction service is performed once per day to generate a 24-h prediction of tomorrow’s power generation using the previous 24-h of meteorological data. The input weather data are retrieved from the data storage of each PV facility.

The second service is anomaly detection on the platform to determine if the sensor connected to the platform is affected by the anomaly. Multiple PV plants are connected to the platform and exchange thousands of communications per minute. In such a scenario, the probability of an anomaly increases, which may impact the platform’s performance by causing a longer delay time, a slower system response, or the loss of multiple data points. When anomalies occur on a platform, the anomaly detection services are directly connected to the message queuing services to provide real-time detection and immediate notification.

#### 3.1.4. User Administration Services

The user administration services include pages and functions for managing user data and configuration. Here, it is possible to configure services such as new user registration, user personal information, user credentials, and user classifications. Typically, the platform supports three distinct user types:Site user: intended for users that are authorized to control only one specific PV plant.Group user: intended for the user to control multiple PV plants that belong to the same group or ownership.Admin user: intended for administrator users who can control and manage all registered PV plants.

#### 3.1.5. PV Plant Administration Services

The administrative data of the PV plant are managed by the PV plant administration service. This service includes registering a new PV plant, editing the PV plant’s detailed information, administering the PV plant’s group, and controlling user access to the PV plant. Then, this service provides functions for registering and managing sensors and devices deployed in a PV plant.

#### 3.1.6. Individual PV Plant Monitoring Services

In addition to monitoring the overall status of the PV plant, it is essential to monitor the status of each individual PV plant for a more complete picture of the system. This service offers to closely monitor the status of the PV plant, administer connected devices and sensors, configure electricity price settings to estimate the PV plant’s revenue, and provide PV plant historical data.

### 3.2. AI Model for PV Power Generation Prediction

Recent advancements in artificial intelligence (AI) have resulted in a broader range of applications. Prediction of time-series data is another common application of AI. AI improves the efficacy of non-linear multi-feature prediction because it is able to better capture the non-linearity between data features. Using weather data as input, an AI model is developed in this work to predict the next day’s PV power generation. Since the platform is connected to multiple PV plants, each with unique characteristics, a single AI model is insufficient for the prediction. Five AI models are created to anticipate the PV power generation of every PV plant. Therefore, each PV plant has five prediction results from each AI model, and the best prediction results among the five AI models developed at that time will be recommended to the user. The following AI models are utilized in this work:

#### 3.2.1. Recurrent Neural Network (RNN)

RNN is a type of neural network designed to process sequential data by implementing a feedback mechanism that allows information to persist over time [21]. The RNN is usually implemented in natural language processing, speech recognition, and time-series data due to its ability to retain messages. The architecture of a typical RNN is shown in Figure 2. The equation for an RNN is as follows:(1)$$h\left(t\right)=\sigma (W_{h,h}h\left(t-1\right)+W_{h,x}x\left(t\right)+b_{h})$$
(2)$$y\left(t\right)=\sigma (W_{y,h}h\left(t\right)+b_{y})$$
where $W_{h,h}$, $W_{h,x}$, and $W_{y,h}$ represents the weight of each node inside the RNN, x(t) is the input data at time *t*, $b_{g}$ and $b_{y}$ represents the bias vector in the model, σ becomes the activation function which in this work, tanh is utilized.

#### 3.2.2. Long Short-Term Memory (LSTM)

LSTM [22] is a special variant of RNN which introduced to overcome weaknesses of the RNN about difficulties to store information over lengthy time intervals. The LSTM has a different structure than RNN to enable learning long-term dependencies. To achieve that, LSTM has memory units to maintain the *cell state* over time. A *gates* is equipped in the LSTM to control the flow of information through the cells. There are four gates in an LSTM, input gate, forget gate, output gate, and cell state gate. In each time *t*, the LSTM receives an input sequence $X=(x_{1},x_{2},\ldots ,x_{t})$ and produces an output sequence of $Y=(y_{1},y_{2},\ldots ,y_{t})$ iteratively from t=1,2,3,…,T where the LSTM updates all units in every time step and followed by calculating loss for all weights. The architecture of LSTM is shown in Figure 3. The equations for LSTM are expressed as follows:(3)$$y_{i}\left(t\right)=\sigma \left(W_{i,x}\left(t\right)+W_{i,j}j\left(t-1\right)+W_{i,s}s\left(t-1\right)+b_{i}\right)$$
(4)$$y_{\varphi }\left(t\right)=\sigma \left(W_{\varphi ,x}x\left(t\right)+W_{\varphi ,j}j\left(t-1\right)+W_{\varphi ,s}s\left(t-1\right)+b_{\varphi }\right)$$
(5)$$Z_{c}\left(t\right)=W_{c,x}x\left(t\right)+W_{c,j}j\left(t-1\right)+b_{c}$$
(6)$$s\left(t\right)=y_{\varphi }\left(t\right)⊙s\left(t-1\right)+y_{i}\left(t\right)⊙g\left(Z_{c}\left(t\right)\right)$$
(7)$$y_{o}\left(t\right)=\sigma \left(W_{o,x}x\left(t\right)+W_{o,j}j\left(t-1\right)+W_{o,s}s\left(t-1\right)+b_{o}\right)$$
(8)$$j\left(t\right)=y_{o}\left(t\right)⊙+h\left(s\left(t\right)\right)$$
(9)$$y_{out}\left(t\right)=f_{out}\left(W_{y_{out,j}}j\left(t\right)+b_{y_{out}}\right)$$
where *Z* denotes the sum of weighted input, *W* represents the weight matrixes, *b* denotes the bias vectors of each gate, and *y* is the output of each gate. Meanwhile, s(t) denotes the current cell state and ⊙ is the element-wise product of vectors. Then, σ, *g*, *h*, and $f_{out}$ is the output activation function which are using tanh in this work. j(t) represents the cell output activation function and $y_{out}\left(t\right)$ is the output of the network.

#### 3.2.3. Bidirectional LSTM (BiLSTM)

BiLSTM is a variant of LSTM that utilizes the forward and backward propagation of information. The overall architecture of BiLSTM is depicted in Figure 4. By utilizing two directions of information, it enables the BiLSTM to better capture the information pattern. The forward layer $l_{f}\left(t\right)$ of a BiLSTM is arranged in an ascending range t=1,2,3,…,T. Meanwhile, the backward layer $l_{b}\left(t\right)$ is arranged in a descending range of t=T,…,1,2,3. The outputs from the forward and backward layers are combined by averaging the outputs. The BiLSTM uses a similar equation as the LSTM with the following modifications:(10)$$l_{f}\left(t\right)=\sigma \left(W_{f_{x,l}}x\left(t\right)+W_{f_{l,l}}l_{f}\left(t-1\right)+l_{f_{h}}\right)$$
(11)$$l_{b}\left(t\right)=\sigma \left(W_{b_{x,l}}x\left(t\right)+W_{b_{l,l}}l_{b}\left(t+1\right)+l_{b_{h}}\right)$$
(12)$$y\left(t\right)=W_{f_{l,y}}l_{f}\left(t\right)+W_{b_{l,y}}l_{b}\left(t\right)+b_{y}$$
where: *W* represents the weight, $b_{y}$ denotes the bias vectors and $i_{f}\left(t\right)$ denotes the forward layer of the model. Meanwhile, $i_{b}\left(t\right)$ represents the backward layer. Then, y(t) is the output of the model.

#### 3.2.4. Convolutional LSTM

The architecture of convolutional LSTM can be seen in Figure 5. The LSTM is only considering temporal information without the ability to process spatial information. A convolutional layer is added to the LSTM to enable understanding spatio-temporal information [23]. The structure of the LSTM used is the same as in regular LSTM. The difference is on the input, where it utilizes a convolutional layer and receives input from spatio-temporal images. The equations for convolutional LSTM are as follows:(13)$$i\left(t\right)=\sigma (W_{x,i}*X\left(t\right)+W_{h,i}*H\left(t-1\right)+W_{c,i}\circ C\left(t-1\right)+b_{i})$$
(14)$$f\left(t\right)=\sigma (W_{x,f}*X\left(t\right)+W_{h,f}*H\left(t-1\right)+W_{c,f}\circ C\left(t-1\right)+b_{f})$$
(15)$$C\left(t\right)=f\left(t\right)\circ C\left(t-1\right)+i\left(t\right)\circ tanh(W_{x,c}*X\left(t\right)+W_{h,c}*H\left(t-1\right)+b_{c})$$
(16)$$o\left(t\right)=\sigma (W_{x,o}*X\left(t\right)+W_{h,o}*H\left(t-1\right)+W_{c,o}\circ C\left(t\right)+b_{o})$$
(17)H(t)=o(t)∘tanh(C(t))
where: *W* denotes the weight of each layer, *b* is the network bias, i(t) represents the input gate, f(t) denotes the forget gate, C(t) is the cell output, and o(t) is the output gate. Meanwhile, H(t) denotes final state, and σ represents the used activation function, which is a tanh or sigmoid in this work.

#### 3.2.5. BiLSTM-Multi Dense

The BiLSTM-Multi Dense model is a model that combines the BiLSTM model with a neural network. The architecture of BiLSTM-Multi Dense is depicted in Figure 6. The model receives a sequence of inputs as in BiLSTM. Meanwhile, the output is a fully connected network (FCN) with multiple layers and neurons. The reason for utilizing an FCN is to better extract information from the sequence output of the BiLSTM. The equation for BiLSTM-Multi Dense is represented by the following equations:(18)$$i\left(t\right)=\sigma (W_{x,i}x\left(t\right)+W_{h,i}h\left(t-1\right)+W_{c,i}\circ C\left(t-1\right)+b_{i})$$
(19)$$f\left(t\right)=\sigma (W_{x,f}x\left(t\right)+W_{h,f}h\left(t-1\right)+W_{c,f}\circ C\left(t-1\right)+b_{f})$$
(20)$$c\left(t\right)=f\left(t\right)\circ c\left(t-1\right)+i\left(t\right)\circ tanh(W_{x,c}x\left(t\right)+W_{h,c}h\left(t-1\right)+b_{c})$$
(21)$$o\left(t\right)=\sigma (W_{x,o}x\left(t\right)+W_{h,o}h\left(t-1\right)+W_{c,o}\circ c\left(t\right)+b_{o})$$
(22)h(t)=o(t)∘tanh(c(t))
where: i(t) represents the input gate, f(t) denotes the forget gate, *b* represents the bias vectors of each gate, c(t) is the cell output and o(t) is the output gate, Meanwhile, h(t) denotes final state.

### 3.3. Isolation Forest for Anomaly Detection

Power plant condition monitoring systems, especially anomaly detection for equipment performance, must be improved to reduce unplanned downtime costs. Moreover, the development of accurate artificial intelligence (AI) models for PV prediction is dependent on high-quality data. Consequently, an increasing need exists for dependable and effective anomaly detection algorithms in this field. Conventional detection techniques that employ fixed thresholds are insufficient for the early identification of anomalies in streaming data from power plants. The inability of these methods to adapt to the data’s constantly changing nature and carry out quick anomaly detection in real time reduces their effectiveness. Consequently, there is a necessity for sophisticated methodologies that can proficiently manage continuous data flow and facilitate the timely identification of irregularities in power plant functionalities. The present research introduces an adaptive threshold isolation forest approach that integrates the Isolation Forest algorithm, a sliding window, and adaptive thresholding.

The isolation forest algorithm [24] is an efficient method for detecting anomalies in computational settings. The data undergoes subsampling and processing within an isolation forest tree structure, which is predicated on random cuts in the values of features that have been randomly selected from the dataset. There is a negative correlation between the depth of sample collection within a tree’s branches and the likelihood of an abnormality, while a positive correlation exists between shorter branch length and the presence of anomalies. The approach employed is capable of quickly recognizing anomalous points from random selections owing to their rarity and uniqueness.

Time-series information has unique characteristics that the isolation forest may not be able to identify, making streaming data anomaly identification challenging. The presence of concept drift or seasonality is a common characteristic in streaming data, posing a major challenge for conventional isolation forests. The occurrence of concept drift signifies that the previous isolation forest model may no longer be dependable in the present patterns and may be inadequate in addressing such circumstances, leading to a decrease in detection performance. As a result, it is essential to begin model retraining and threshold updates based on incoming data to ensure sustained detection performance.

An adaptive threshold isolation forest is utilized for detecting anomaly points in streaming data, following the procedure described in Algorithm 1. The isolation forest algorithm functions as an anomaly detection technique by transforming incoming time-series data points into outlier scores. The adaptive threshold functions as a classifier for determining the level of normality or abnormality of individual data points. The sliding window technique proficiently handles the occurrence of concept drift or seasonality in streaming data by periodically retraining the isolation forest model in real time. Our methodology guarantees the dependability of the anomaly detection mechanism in dynamic settings while maintaining the imperative of minimal system memory consumption.
**Algorithm 1** Adaptive Threshold Isolation Forest for Anomaly Detection in Streaming Data**Input: ***D*—the streaming data (at time *t*)            *W*—the size of the sliding window            *N*—a factor used to calculate the adaptive threshold            *I*—the frequency of updating the model            *R*—the threshold for contamination ratio or the anomaly rate**Output: ***A*—a list of indices where anomalies were detected1:Initialize anomaly array A= empty.2:Initialize buffer B= empty array.3:Initialize model with pre-train Isolation Forest model.4:Set the counter(c), current index(idx), contamination ratio(r) to zero.5:Set the threshold =0.6:**while** True **do**7:    Append incoming data *D* to buffer *B*.8:    Predict anomaly score of $D_{t}$ using model.9:    **if** anomaly score≥adaptive threshold **then**10:        Append incoming data *D* to anomaly array *A*.11:        r←lengthofanomalyarray/lengthofbuffer12:    **end if**13:    **if** r>R **then**14:        start index=max(0,idx−W) & end index$=\min(\mathrm{length}\;\mathrm{of}\;B_{t},idx+W)$.15:        Calculate the mean & standard deviation of $B_{t}$ from start to  endindex.16:        Adaptive threshold←mean+N∗standard deviation.17:        **if** $\mathrm{length}\;\mathrm{of}\;(B_{t})\geq W$ **then**18:           **if**
*c* = *I*
**then**19:               Re-train Isolation Forest model $\leftarrow IF(r,B_{t})$.20:               model← re-trained Isolation Forest model.21:               c=0.22:           **end if**23:           Remove the oldest data point from the buffer.24:        **end if**25:        c←c+1.26:    **end if**27:    idx←idx+1.28:**end while**

Setting the counter, index, and contamination ratio to zero is the first step in the procedure before initializing an already-existing isolation forest model. Furthermore, an empty array is created and referred to as the buffer, which functions as a temporary storage for the data being streamed. As new data arrives, it is appended to the buffer, and the Isolation Forest model is used to predict the anomaly score. In the event that the anomaly score surpasses the adaptive threshold for a given index, it is appended to the anomaly array, and the contamination ratio is recalculated. Furthermore, in cases where the cumulative data stored in the buffer exceeds the predetermined window size and the counter exceeds the designated update frequency, it becomes necessary to retrain the isolation forest. This is achieved by utilizing the latest contamination ratio and optimizing the buffer size to ensure optimal performance. Therefore, this approach efficiently mitigates the issue of concept drift while preserving minimal memory demands.

## 4. Results and Discussion

### 4.1. Intelligent IoT Platform Development

The intelligent IoT platform is built using Flask as the backend framework. The backend is the orchestrator that synchronizes all of the platform’s components. The infrastructure resides and is executed on a cloud server. To serve the client, HTML and JavaScript are used to create a user interface (UI) page that can exchange data with the backend using WebSocket and REST API technologies. Because the platform is hosted on a cloud server, users can access it from anywhere.

The intelligent IoT platform deploys MQTT, which functions as message queuing, on a cloud server to manage all message exchange between clients and servers. Quality of service (QoS) 1 is selected in order to achieve a balance between system reliability and efficiency. QoS 1 ensures that the message is delivered to the receiver at least once, while the transmission time is relatively short to facilitate a faster system response.

To evaluate the efficacy of the intelligent IoT platform, the system’s packet error rate during message exchange is measured. Thirty-eight clients are connected to the intelligent IoT platform developed for the testing environment. Each client transmits a total of five messages to the edge server, which then forwards them to the cloud server. In this test, each client message is formatted as a 500-byte String message. Using a cron job scheduler, every client transmits messages simultaneously. To test the efficacy of the system, the message transmission procedure is repeated 200 times. Then, the received messages on the cloud server are evaluated, where the number of received messages and the average message transmission time are computed. Table 1 displays the results of the test.

From Table 1, it can be seen that during the test, a total of 238,000 messages were transmitted from the clients. The total messages are coming from each of the 38 clients, where each client transmits five messages and repeats them 200 times. On the cloud server, it is recorded that 238,000 messages were received at the end of the testing. Hence, all messages transmitted by the clients are well received by the cloud server, which results in a 0% error rate in data transmission. Then, the average transmission time measures the time required for each message to arrive at the cloud server after being transmitted from each client. The average transmission time is measured at 0.142 s, which is quite fast considering that the message needs to go to the edge server first before arriving at the cloud server. This result also proves that MQTT QoS 1 guarantees that the message reaches the receiver at least once within a relatively short time.

The management pages for the IoT platform are divided into two groups of web pages. The first set of pages is designed to serve admin and group users, allowing them to monitor all PV plants connected to the platform and the system’s overall summary. The second set of pages is designed specifically for the user and only displays information from individual PV facilities. The admin and group users can access the second set of pages to administer individual PV plants from the first set.

Figure 7 shows six important pages of the developed IoT platform, where Figure 7a demonstrates the dashboard page for admin and group users to enable all PV plant monitoring, control, and configuration. All PV plants accumulated PV power generation can be monitored in Figure 7b where it shows the PV power generation data from individual PV plants and regional. The pages for monitoring anomaly detection in all PV plants are shown in Figure 7c, where the detected sensor anomaly will be marked as a sensor breakdown that needs additional attention by the users to further analyze the issue.

In Figure 7d, the dashboard for the second set of pages is shown, whose function is to provide a dashboard for the management and control of individual PV plants. To monitor the real-time status of individual PV plants, Figure 7e is utilized, where the pages contain a summary of the current PV plant status and information. In each PV plant, PV power generation prediction is performed where it can be accessed, as shown in Figure 7f.

### 4.2. PV Power Generation Prediction Performance

Five AI models have been developed to predict PV power generation. Since all connected PV plants are located in Korea, all five models incorporate meteorological data collected by the Korean Meteorological Administration (KMA) for each PV plant location. Using the provided API, the KMA provides meteorological data from hundreds of weather stations in Korea. By having the address information of each PV plant, the corresponding weather data can be collected from the KMA.

In this work, we collected the measured weather data from the weather stations. A total of seven data features are collected to be modeled and generate the PV power generation prediction. Those seven data points are temperature, humidity, wind speed in *x* direction, wind speed in *y* direction, intensity of solar radiation, cloud density, and duration of sunshine. The hourly weather data are collected from the KMA from 2021 to 2023. To train the AI model, target data are required for the PV power generation from each PV plant, with data dates ranging from 2021 to 2023.

Each PV plant generates various amounts of energy due to its unique location. To train an AI model for predicting PV power generation, a distinct set of input and output data is required. Algorithm 2 demonstrates how to train the AI model on the platform:
**Algorithm 2** Method to train AI models on multiple PV plants**Input: ***M*—The list of the AI model            *P*—list of monitored PV plant            *W*—weather dataset for each PV plant            PG—generated power of each PV plant**Output:**MW—a set of trained model weights for each PV plant1:**while** True **do**2:    **for** model in M **do**3:        **for** plant in P **do**4:           load the plant’s *W* as model input.5:           load the plant’s PG as model target.6:           train the model using the input and target data.7:           save the trained model weights to a list of MW.8:        **end for**9:    **end for**10:**end while**

Algorithm 2 explains that the AI model training to generate the next day’s power generation is sequentially conducted for each PV plant connected to the platform. Each PV plant should develop five AI models using a serial training schedule. For each AI model training, each PV plant requires a meteorological dataset as input data and a generated power dataset as target data. The training procedure for each PV plant is carried out on a cloud server with sufficient computing capacity. After training, a list of model weights for each AI model and PV plant are received, and each model weight is transmitted to the edge server of each PV plant in order to determine the next day’s power generation.

In Algorithm 3, the steps to train each AI model are outlined, with each AI model receiving seven input features of hourly weather observation data with a 24-h data length as the input. Meanwhile, the generated PV power data are utilized for the target data, where the duration is 24 h of future hourly data. The input and target data are pre-processed prior to training the model, such as by normalization, scaling, and data windowing. Once the data are prepared, the model is trained and evaluated until the desired error value is reached. The trained model weights with the lowest error value are then preserved as the optimal model weights to be used again for inference on the PV plant’s edge server. After training the AI model using the methods described in Algorithms 2 and 3, the training results for each trained AI model can be derived and are displayed in Table 2.
**Algorithm 3** Method to train individual AI models**Input:** Temperature as input data            Humidity as input data            Wind speed in *x* direction as input data            Wind speed in *y* direction as input data            Solar radiation intensity as input data            Cloud density as input data            Duration of sunshine as input data            Generated PV power as target data            Model structure and hyperparameter**Output:** Predicted 24-h ahead PV power generation1:load the input data.2:load the target data.3:normalize the input and target data.4:scale the input and target data into −1 to 1.5:form the input and target data into a window with input sequence length and target sequence length as 24.6:divide the data into batches.7:**while** model error > target error **do**8:    **for** data in batches **do**9:        train the model using the data.10:        evaluate the model.11:    **end for**12:**end while**13:save the best model weights with lowest error value.

In Table 2, the performance of five AI models is evaluated using four evaluation metrics: MSE, MAPE, MAE, and $R^{2}$. The MSE and MAE are employed to measure the quality of the prediction results. Meanwhile, MAPE and $R^{2}$ are used to evaluate the prediction accuracy and variations in the prediction results, respectively. The BiLSTM is clearly seen as the best-performing model because it achieves the best value on all evaluation metrics. The predicted value from BiLSTM is closely similar to the ground truth data, measured by the MSE and MAE values. The BiLSTM also achieves the most accurate model with the lowest MAPE value. The $R^{2}$ value of BiLSTM is the highest among other models, which explains why BiLSTM was able to generate a good fit with the data.

Figure 8 depicts the predicted 24-h PV power generation for all AI models. The BiLSTM prediction closely matches the observed trend. BiLSTM is followed by BiLSTM-Multi Dense, which has marginally inferior prediction performance. LSTM and convolutional LSTM then perform worse than BiLSTM-Multi Dense in terms of prediction accuracy. In addition, RNN becomes the model with the worst performance among the five developed AI models, as its prediction trends and value are inferior to the ground truth data.

As a result, the BiLSTM model is used as the primary model in the platform to predict the next day’s power generation for each PV facility. However, the stochastic nature of PV power generation may diminish BiLSTM’s performance. Algorithm 2 can be regarded as an algorithm for AI model re-training when the prediction performance of the current AI model degrades. The purpose of the re-training method is to select the new AI model with the highest performance for use on the platform, thereby preserving the outstanding performance of power generation prediction. On the platform, the user can access the prediction results from other models in order to conduct comparisons and cross-validation.

### 4.3. Anomaly Detection Performance

In this study, a dataset collected by an environment sensor installed at the PV site is used to evaluate the proposed method for detecting irregularities using anomaly detection. The dataset was structured on purpose to include regulated anomalies. The detailed information regarding this dataset can be found in Table 3. Figure 9 depicts the anomaly detection outcome of our adaptive threshold isolation forest method. Figure 9a depicts the temporal sequence as well as the anomalous point, identified as an extreme point anomaly and highlighted in orange windows. The results depicted in Figure 9b demonstrate the efficacy of our method for identifying anomalies through the use of anomaly scores, which exhibit a clear distinction between normal and abnormal data. As observed, the performance of the proposed method demonstrates a high degree of precision when identifying outliers in the entire dataset. Using the scoring mechanism of the isolation forest model, the aforementioned methodology facilitates the identification of anomalous sequences.

Table 4 presents an evaluation of the performance of different lightweight data anomaly detection techniques in time-series data, based on precision, recall, and training time. Along with our proposed method, the comparison includes HBOS, KNN, OCSVM, and LOF techniques. The concept of precision in anomaly detection refers to the ratio of accurately classified positive anomalies to the total number of positive anomalies. On the other hand, recall refers to the ratio of accurately classified positive anomalies to the overall number of actual positive anomalies present in the dataset. The training duration is utilized as a metric to assess the complexity of the model. The findings show that the suggested approach successfully achieves a training duration of 3.318 s and produces noteworthy levels of precision (0.9517) and recall (0.998). The HBOS algorithm achieves a shorter training duration of 2.113 s, albeit with a slightly diminished precision score of 0.822 and recall score of 0.969. The LOF algorithm demonstrates a lower training time of 1.182 s, although this comes at the expense of diminished precision (0.672) and recall (0.539). The MCD model demonstrates a notable training efficiency of 0.04 s while also achieving a satisfactory level of precision (0.824) and recall (0.769). The OCSVM algorithm is characterized by a significantly prolonged training duration of 209.499 s. Nevertheless, it showcases exceptional precision and recall performance metrics, with values of 0.920 and 0.999, respectively. In general, the proposed strategy achieves an optimal balance of training duration and performance as measured by precision and recall. The training duration can also be optimized using a multi-threading scenario.

In the proposed model for anomaly detection in solar power generation, false positives and false negatives are important considerations. False positives refer to cases where the system incorrectly detects an anomaly when there is none, while false negatives occur when the system fails to identify an actual anomaly. To address false positives, the model incorporates a multi-level validation mechanism. The initial anomaly detection is performed by an adaptive threshold Isolation Forest model trained on historical data and patterns. However, to minimize false positives, the system employs a secondary validation step using cross-validation techniques monitored by the expert. This step helps to filter out potential false positives by examining the consistency and reliability of anomaly indicators across multiple photovoltaic installations. Furthermore, the method in Algorithm 1 also incorporates a feedback loop for contamination ratio, where the system continuously learns from feedback on its detections. This allows the proposed method to improve over time by adapting to specific installation conditions and reducing both false positives and false negatives. In the case of false negatives or missed alarms, the system leverages real-time monitoring and predictive modeling to enhance sensitivity. By continuously analyzing and learning from incoming data, the proposed method can adjust its anomaly detection thresholds and patterns to improve the accuracy of identifying anomalies. Overall, the proposed architecture aims to strike a balance between minimizing false positives and false negatives through multi-level validation, continuous learning, and adaptive mechanisms.

The anomaly detection ratio also has a positive correlation with the power generation prediction, suggesting that a smaller anomaly detection ratio can lead to improved prediction performance. The anomaly detection ratio provides valuable feedback for the day-ahead prediction capabilities. Anomalies detected during monitoring can help refine and improve power generation predictive models by incorporating the observed behavior. This feedback loop allows the predictive models to adapt and account for previously unseen patterns, enhancing their accuracy and reliability in future predictions. Furthermore, by incorporating labeled anomaly data into the training process, the models can capture a wider range of scenarios, including both normal and abnormal conditions.

## 5. Conclusions

This work develops an intelligent IoT platform for monitoring multiple PV facilities. The intelligent IoT platform is utilized to monitor diversely located PV facilities in Korea. Multiple components are utilized to support the platform’s functionalities. The platform collects and transmits each PV plant’s data to the cloud server via an edge server. Using an edge server reduces the platform’s latency, resulting in a quicker system response time.

This work proposes a method for predicting the next day’s photovoltaic (PV) power generation based on the previous 24 h of meteorological data. Five AI models are used to make the prediction, with the BiLSTM model achieving the highest performance. BiLSTM becomes the primary AI model for predicting the next day’s power generation, with other models’ predictions being available for comparison and cross-validation.

On the platform, anomaly detection was performed using an adaptive threshold isolation forest to detect anomalies in the PV plant. By having an anomaly detection system monitor anomalies in the sensors of each PV plant, the user of the platform is able to detect PV plant errors early on. The results demonstrate that adaptive threshold isolation forests can effectively detect sensor anomalies.

We intend to implement anomaly detection in the larger implementation area of the PV plant in future work to enable more precise fault detection in the PV plant. Additional experiments and modifications to the PV plant are required to perform these tasks. Also anticipated is an upgrade to the platform’s architecture, in which the backend’s infrastructure may be replaced with a more efficient one. The message queuing system will also be converted to more modern technology, such as Apache Kafka, to allow for a greater number of connected devices on the platform while maintaining a quick response time.

## Figures and Tables

**Figure 1 sensors-23-06674-f001:**
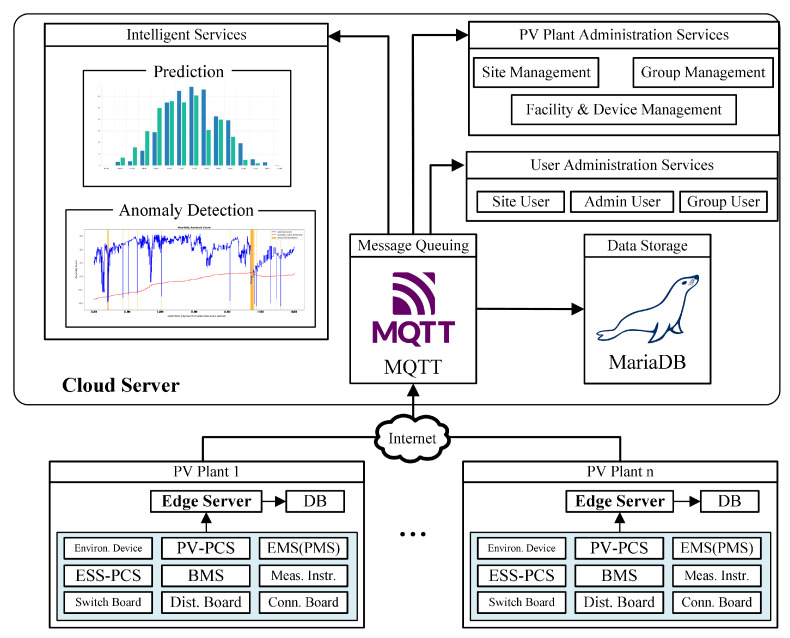
The proposed intelligent IoT platform for multiple PV plant monitoring.

**Figure 2 sensors-23-06674-f002:**
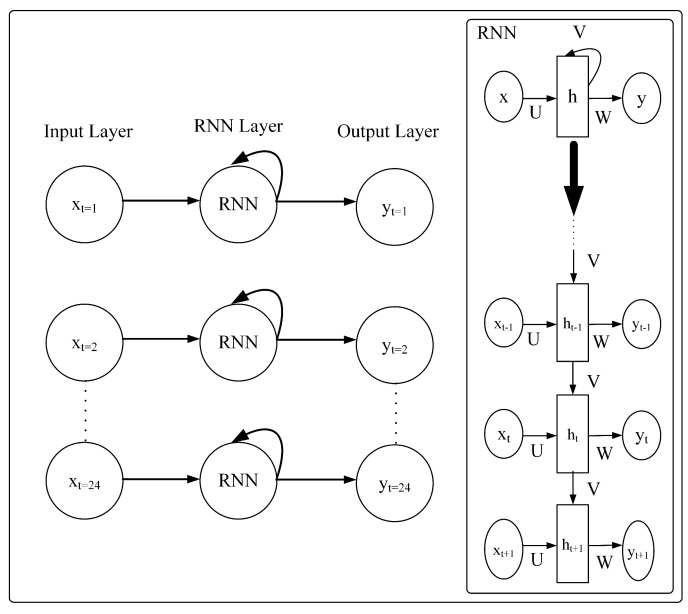
RNN Architecture.

**Figure 3 sensors-23-06674-f003:**
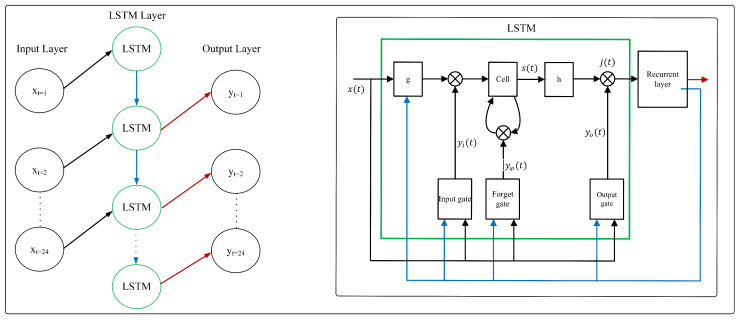
LSTM Architecture.

**Figure 4 sensors-23-06674-f004:**
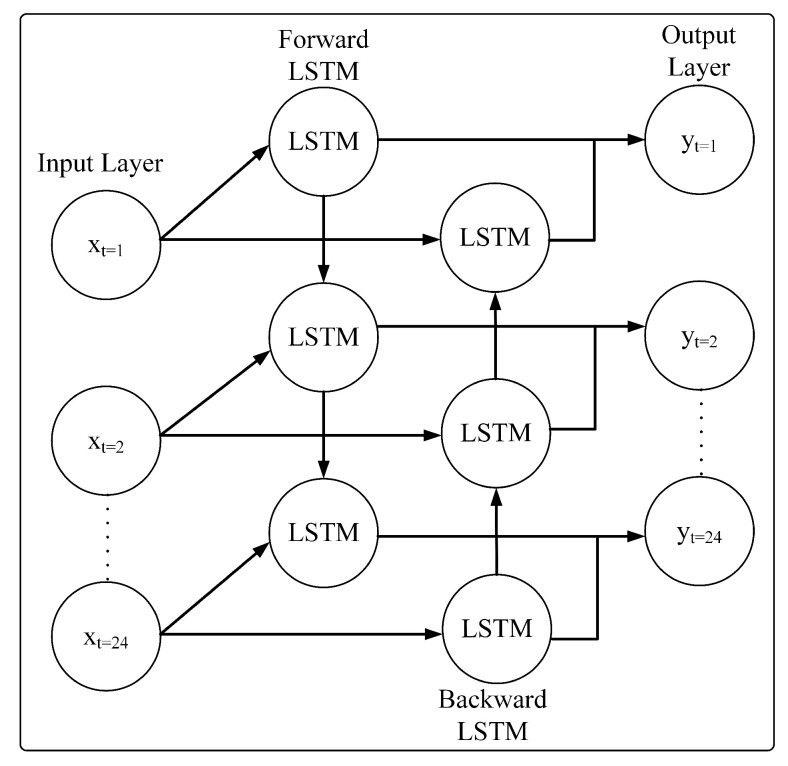
BiLSTM Architecture.

**Figure 5 sensors-23-06674-f005:**
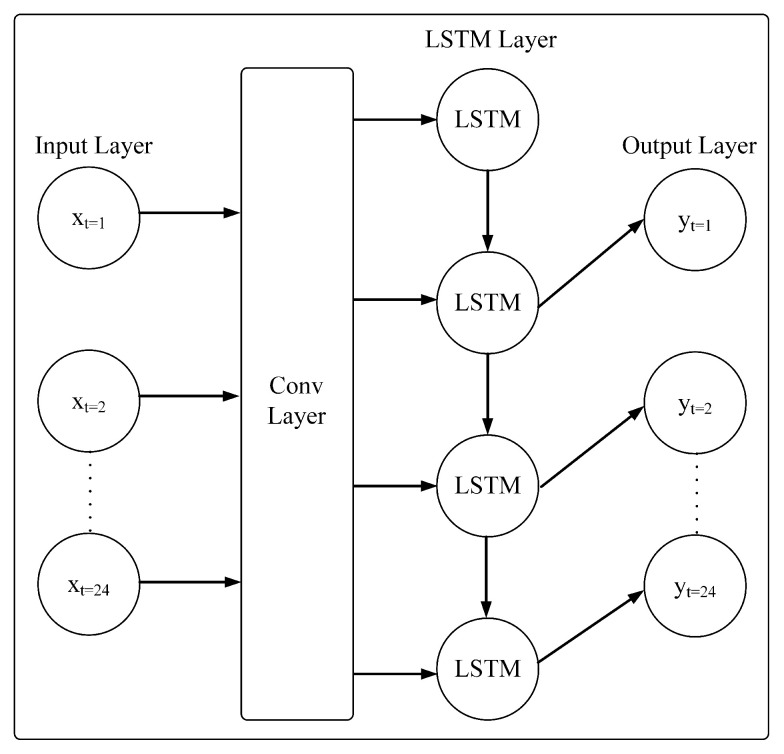
Convolutional LSTM Architecture.

**Figure 6 sensors-23-06674-f006:**
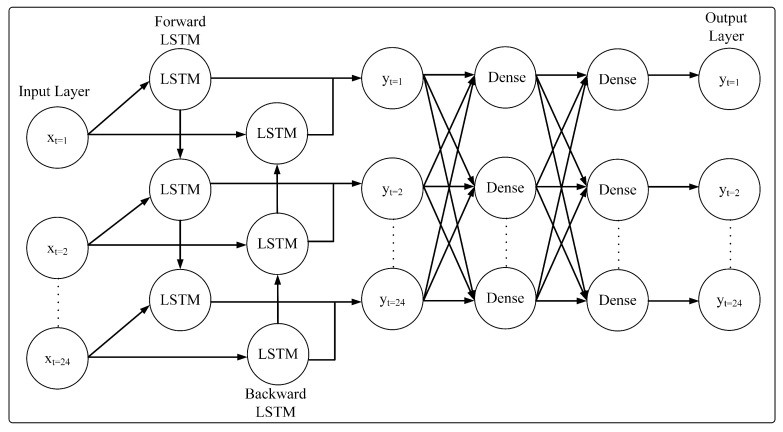
BiLSTM-MultiDense Architecture.

**Figure 7 sensors-23-06674-f007:**
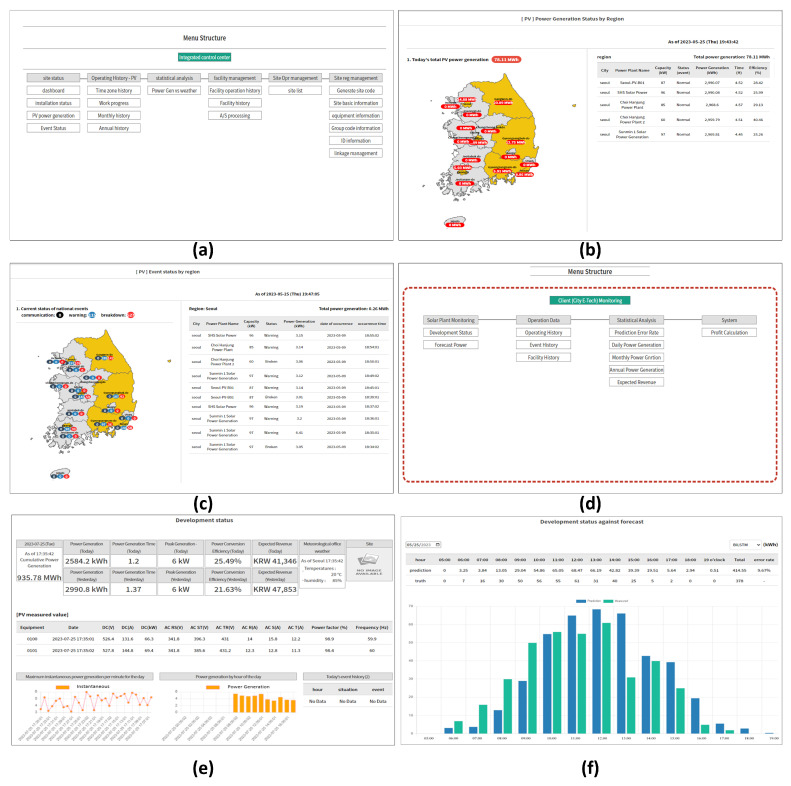
The website pages for multiple PV plant monitoring. (**a**) Dashboard page for all PV plants monitoring; (**b**) All PV plants accumulated power generation monitoring page; (**c**) All PV plants anomaly detection monitoring page; (**d**) Dashboard for individual PV plant monitoring page; (**e**) Individual PV plant summary page; (**f**) Next day PV power generation prediction page.

**Figure 8 sensors-23-06674-f008:**
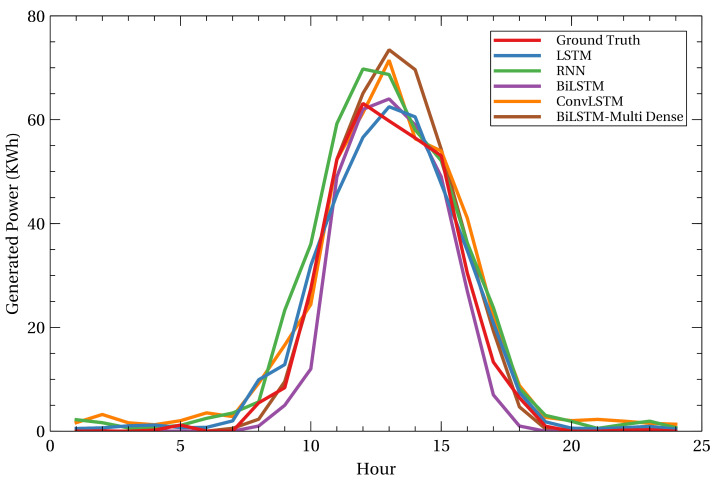
Comparison of the next 24-hour PV power generation prediction results from five AI models.

**Figure 9 sensors-23-06674-f009:**
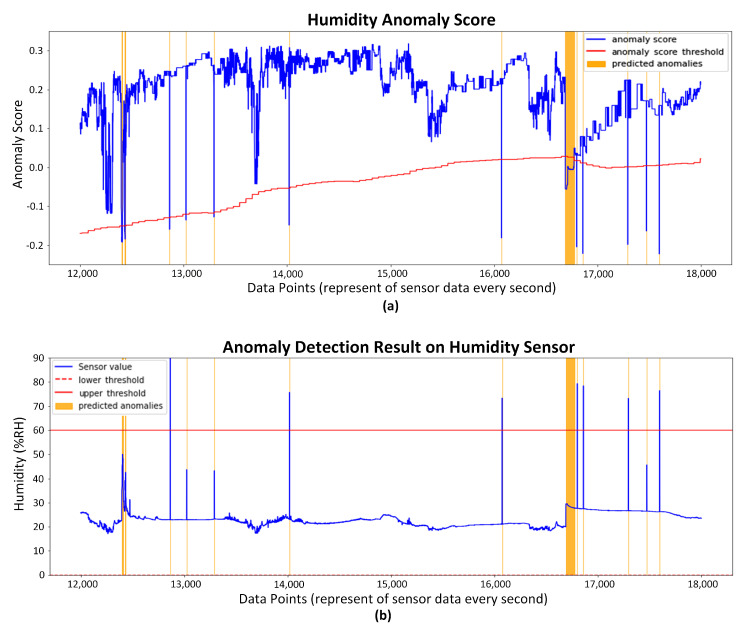
Anomaly detection by our proposed method for humidity data in PV site. (**a**) Anomaly points with original humidity measurement; (**b**) Anomaly score with adaptive threshold.

**Table 1 sensors-23-06674-t001:** Results of packet error rate evaluation.

Number of Clients	Message Size (bytes)	Number of Message	Total Transmitted Messages	Average Transmission Time (s)	Total Received Messages	Error Rate (%)
38	500	5	238,000	0.142	238,000	0

**Table 2 sensors-23-06674-t002:** Training results from each trained AI models.

AI Model	MSE	MAPE	MAE	$R^{2}$
RNN	0.0272	0.3152	0.1084	0.8866
LSTM	0.0152	0.2334	0.0755	0.9215
BiLSTM	**0.0072**	**0.1982**	**0.0542**	**0.9664**
Convolutional LSTM	0.0172	0.2974	0.0980	0.9202
BiLSTM-Multi dense	0.0125	0.2243	0.0662	0.9526

Note: the best performance for each metric in **bold**.

**Table 3 sensors-23-06674-t003:** Detailed information of labeled dataset.

Data Split	Total Sample	Normal	Anomaly
Training	39,345	38,976	369
Testing	20,000	19,814	186

**Table 4 sensors-23-06674-t004:** A comparison of the proposed anomaly detection method and other techniques.

Anomaly Detection Techniques	Training Time (s)	Precision	Recall
Proposed method	3.318	**0.9517**	0.998
HBOS	2.113	0.822	0.969
LOF	1.182	0.672	0.539
MCD	**0.04**	0.824	0.769
OCSVM	209.499	0.920	**0.999**

Note: the best performance for each metric in **bold**.

## Data Availability

The data presented in this study are available on request from the corresponding author. The data are not publicly available due to further research on processing.

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
