# Peer review of "Intelligent IoT Platform for Multiple PV Plant Monitoring"

_sensors, 2023, doi:10.3390/s23156674_

Round 1
Reviewer 1 Report
This manuscript’s goal is to develop a smart IoT-enabled monitoring system for photovoltaic (PV) panels. The authors used a variety of models, predicting the next day’s (PV) power generation based on the previous 24 hours of meteorological data. Moreover, they proposed real-time anomaly detection to enhance the effectiveness of the developed IoT platform.
The trained models were also checked and evaluated using several assessment metrics, including mean squared errors, root mean squared errors, and coefficient of determination.
This manuscript is well-written and well-organized. The study's objectives are clearly stated, and the results of the research are well-examined. However, there are some typo errors and minor drawbacks including the following:
· Page 1 line 25 1.73x105 the x should be replaced by multiplication operator.
· Regarding the used MQTT protocol, the authors did not mention which quality of service they selected. Also, they did not evaluate the packet error rate of received data.
· The Quality of Figures 7 and 9 needs to be improved.
Author Response
Dear Reviewer,
We would like to thank you for the comments and the opportunity to resubmit a revised version.
We have updated our manuscript according to your advice and highlighted it in yellow color.
Please see the attachment.
Thank you very much.
Best regards,
Professor Yeong Min Jang
Department of Electronics Engineering,
Kookmin University, Seoul, Korea.

Reviewer 2 Report
The article by Utama et al. presents a score of artificial intelligences to predict solar power generation and to detect anomalies for monitoring multiple photovoltaic installations at the same time. The article is well written, interesting, and significant, therefore, I recommend publication in Sensors. Below, I provide some comments which I think the authors should address before publication:
I am missing some commenting on the possibility of false positives in anomaly detection and what to do with them.
Is there a cross influence between the 24h prediction capabilities and the anomaly detection ratio? Can they benefit from each other in the performance of each of the AI models?
Minor language editing would benefit the article, specially on the abstract and introduction.
Author Response

(The authors gave the same response as above.)
